# The Clinical Implications of Progesterone in Preeclampsia

**DOI:** 10.3390/biom15101458

**Published:** 2025-10-15

**Authors:** Zhenzhen Liu, Weirong Gu

**Affiliations:** 1Key Lab of Reproduction and Development, Department of Obstetrics and Gynecology, Obstetrics and Gynecology Hospital of Fudan University, Shanghai 200011, China; 21211250010@m.fudan.edu.cn; 2Shanghai Key Laboratory of Female Reproductive Endocrine Related Diseases, Shanghai 200011, China

**Keywords:** preeclampsia, progesterone, vascular function, immune response, placental function, clinical application

## Abstract

Preeclampsia is a severe complication affecting both maternal and neonatal health and is becoming a significant global public health issue. As a vital steroid hormone, progesterone (P4) plays a crucial role during pregnancy and in regulating various physiological processes. Recent studies have indicated that P4 is not only involved in pregnancy maintenance, but may also be closely related to preeclampsia pathogenesis and prevention. Previous research has suggested that P4 may participate in the mechanism of preeclampsia by regulating vascular function, immune responses, and placental function. Moreover, key enzymes and metabolites involved in the synthesis and metabolism of P4 are also associated with preeclampsia onset. Additionally, the potential value of clinically applying P4 in preventing and treating preeclampsia has been shown; however, the corresponding clinical practices require further validation and optimization. This study aimed to review the physiological effects, pathological functions, and clinical applications of P4 in preeclampsia, providing evidence for future research and clinical practice.

## 1. Introduction

Preeclampsia (PE) is defined as the onset of new hypertension after 20 weeks of gestation with or without proteinuria. In severe cases, preeclampsia may also be associated with renal, cardiac, pulmonary, hepatic, and neurological dysfunction, as well as hematological disorders, fetal growth restriction (FGR), stillbirth, and even maternal mortality [1]. PE affects 2–8% of pregnant women worldwide and is a major cause of morbidity and mortality for both mothers and fetuses; however, the only known cure for PE is delivery [2,3].

Progesterone (P4) is a cholesterol-derived hormone that is essential for establishing and maintaining pregnancy [4]. A systematic review and meta-analysis showed that the use of vaginal micronized P4 in the first trimester may reduce the risk of hypertensive disorders of pregnancy (HDPs) and PE in singleton pregnancies [5,6]. The oral P4 preparation dydrogesterone significantly reduced PE incidence in women with higher-risk pregnancies and reduced hypertension, proteinuria, FGR, and preterm labor when supplemented in the first and second trimester (from 6 to 20 weeks of gestation) [7,8]. This evidence implies the important role of P4 in preventing PE, the mechanism of which is crucial to explore.

This review aims to characterize the potential mechanisms by which P4 is involved in PE, including its effects on vascular function, immune response, and placental function. By regulating P4 levels, the key enzymes and metabolites involved in P4 synthesis and metabolism are also associated with PE onset. In addition, we discuss the application potential of P4 in clinical management and PE-like animal models. Through a comprehensive analysis of the literature, we hope to provide new insights and evidence regarding the mechanisms, prevention, and treatment of PE, thereby improving maternal and infant health.

## 2. P4 Synthesis and Metabolism

### 2.1. P4 Synthesis Process

As a steroid hormone synthesized by the ovarian corpus luteum, placenta, and adrenal glands, P4 plays a crucial role during pregnancy and other physiological activities [9]. Its synthesis requires the cytochrome P450 side-chain cleavage enzyme (P450scc) to convert cholesterol inside the mitochondria to pregnenolone (PG), followed by the conversion of PG to P4 by 3β-hydroxysteroid dehydrogenase (3β-HSD) in the endoplasmic reticulum. The synthesis of P4 and its key enzymes is illustrated in Figure 1. Synthesized P4 can be released into the extracellular space through simple diffusion before it enters the bloodstream and exerts its physiological effects.

### 2.2. Key Enzyme or Metabolite Abnormalities During P4 Synthesis in PE

Abnormalities in the key enzymes or metabolites during P4 synthesis are closely related to PE (Figure 1). Cholesterol is the raw material in progesterone synthesis. Cellular cholesterol homeostasis is differentially regulated in extravillous trophoblast cells (EVTs) compared with villous cytotrophoblast cells (VCTs) using primary human trophoblasts isolated from placental tissues in the first trimester [10]. EVTs display increased levels of free and esterified cholesterol. Additionally, PG concentrations tended to decrease in the serum and placentas of PE [11], as well as in cellular PE models induced by hypoxia and N nitro L arginine methyl ester hydrochloride (L-NAME) [12]. Moreover, 3β-HSD levels are decreased in EVTs of idiopathic recurrent spontaneous abortions [10], while its expression is decreased in cellular hypoxia-induced PE models [12]. Maternal cadmium exposure contributes to PE onset in pregnant rats by disrupting local P4 synthesis in the placenta and inhibiting 3β-HSD expression [13]. Additionally, it has been reported that the expression of cytochrome P450 family 11 subfamily A member 1 (CYP11A) is significantly increased in placentas of severe PE (sPE) compared with normal pregnancy. An abnormally high expression of CYP11A inhibits trophoblastic proliferation, triggers excessive mitochondrial oxidative stress, and increases apoptosis [14,15]. CYP11A expression was increased in cellular PE models induced by hypoxia, L-NAME, and catechol-O-methyltransferase inhibitor (COMT-1) [12]. CYP11A1 overexpression in rat pregnancies leads to PE-like symptoms [16]. This evidence indicates that abnormalities in key enzymes or metabolites during P4 synthesis are closely related to PE.

### 2.3. P4 Synthesis Regulation

P4 synthesis and secretion are regulated by various factors, including the action of luteinizing hormone (LH) prior to ovulation, which prompts the ovarian corpus luteum to synthesize a substantial amount of P4 [17]. If the egg is successfully fertilized, the ovarian corpus luteum transforms into a pregnancy corpus luteum, which, under the influence of human chorionic gonadotropin (hCG) secreted by the placenta, maintains its function and continues to secrete P4. At approximately 10 weeks of gestation, the placenta begins to synthesize P4 on its own to support pregnancy and ensure proper embryonic development.

### 2.4. hCG Abnormalities in PE

Abnormalities in hCG may be related to PE development (Figure 1). The serum hCG concentration is significantly increased in patients with PE and is associated with its severity in early, mid-, and late pregnancy. Additionally, in early pregnancy, the hCG concentration can be used as a marker to predict PE risk with better sensitivity and specificity [18,19,20,21,22]. It has been reported that every sigma (standard deviation) increase in β-hCG level leads to a 2.73-fold increase in PE risk [23,24,25,26,27,28,29]. This evidence shows that abnormal P4 synthesis regulation may also be involved in PE pathogenesis.

### 2.5. P4 Metabolism

P4 metabolism mainly occurs in the liver, where it is converted into water-soluble metabolites through reduction, hydroxylation, and conjugation reactions for excretion (Figure 2) [30,31]. First, P4 is catalyzed by 5α-reductase or 5β-reductase to form dihydroprogesterone (DHP), which is then converted by 3α-hydroxysteroid dehydrogenase (3α-HSD) into pregnanedione and further reduced by 20α-hydroxysteroid dehydrogenase (20α-HSD) to yield pregnanetriol. In the alternative pathway, P4 undergoes hydroxylation by 17α-hydroxylase (CYP17A1) to generate 17α-hydroxyprogesterone (17-OHP). Additionally, a small portion of P4 is hydroxylated by CYP450 enzymes, such as aromatase (CYP19A1), to form less active metabolites, such as 6β-hydroxyprogesterone or 16α-hydroxyprogesterone. Finally, these metabolites (e.g., pregnanetriol and pregnanedione) are conjugated with glucuronic acid or sulfate via UDP-glucuronosyltransferase (UGT) or sulfotransferase (SULT), forming water-soluble conjugates, such as pregnanediol glucuronide (P3G), which are primarily excreted in urine or bile. Individual variations in metabolic pathways are influenced by genetics, liver and kidney function, and hormonal status (e.g., pregnancy).

### 2.6. Key Enzyme or Metabolite Abnormalities During P4 Metabolism in PE

The abnormal expression of key enzymes or metabolites in P4 metabolism is associated with PE (Figure 2). Research has shown that the precursor-to-product ratio of 5α-reductase is significantly increased in the serum of women with PE [32]. 5α-reductase type 1-deficient mice exhibit impaired decidualization and disrupted angiogenesis signaling pathways [33]. Additionally, it has been reported that the precursor-to-product ratios of CYP17A1 were significantly decreased in the serum of women with PE [32]. The expression of CYP17A1 was significantly decreased in a hypoxia-induced cellular PE model [12]. Interestingly, genetic polymorphisms in CYP17A1 might play a crucial role in PE, especially in early-onset sPE [34]. For example, CYP17A1 rs4919690 and rs4919687 are associated with an increased risk of sPE [34], while CYP17A1 rs1004467 and rs3824755 seem to be closely associated with mild PE in Han women [35]. Furthermore, research has shown that CYP19A1 expression is relatively low in the placentas of PE patients [36]. Animal experiments have shown that low CYP19A1 expression in rat pregnancies leads to PE-like symptoms [36]. Research has found that low CYP19A1 expression enhances the invasion and migration abilities of trophoblasts [36]. In addition, 17-OHP levels were significantly higher in the serum of patients with PE than in normal women [37]. A randomized open-label controlled study showed that 17-OHP has no effect on improving maternal or neonatal outcomes in conservatively managed early-onset PE (EOPE); however, it does alter the levels of inflammatory markers such as interleukin-6 (IL-6) and tumor necrosis factor-alpha (TNF-α), which could reduce PE pathogenesis [38]. Animal experiments have shown that the intraperitoneal administration of 17-hydroxyprogesterone caproate (17-OHPC) can reduce hypertension, fetal demise, T cells, and NK cells in response to placental ischemia in PE-like rat models induced by reduced uterine perfusion pressure (RUPP), soluble fms-like tyrosine kinase-1 (sFlt-1), and TNF-α [39,40,41,42,43]. This evidence indicates a relationship between the key P4 metabolism enzymes or metabolites and PE.

In the above, we reviewed the P4 synthesis and metabolism processes, as well as the key enzymes and metabolic products involved, and their correlation with PE pathogenesis; this was validated using cell, animal, and clinical specimens. This evidence suggests that P4 may play an important role in PE pathogenesis and may be involved in its early prediction and prevention. Next, we summarize the preventive effects of P4 itself on PE, as well as the animal model studies and clinical research.

## 3. The Potential of P4 Supplementation in Preventing PE

### 3.1. Animal Research

Animal experiments show that P4 levels were lower in the urine and blood of PE-like models [11,44,45]. In addition, P4 can prevent PE-like symptoms in several animal models induced by L-NAME, sirtuin 1 (SIRT1) knockdown, RUPP, and cadmium (Table 1) [4,41,44,46]. Additionally, progesterone-induced blocking factor (PIBF) can reduce hypertension, inflammation, and fetal growth restriction in a RUPP rat model of PE [47], while PIBF blockade causes hypertension, inflammation, and signs of endothelial dysfunction, all of which are associated with PE [48]. Furthermore, impaired luteal phase P4 signaling via the administration of the P4 antagonist RU486 (mifepristone) causes fetal loss and FGR during late gestation [49]. This evidence demonstrates that P4 supplementation during pregnancy might play a role in PE treatment and prevention and may be involved in its pathogenesis.

### 3.2. Clinical Research

Insufficient P4 leads to PE development, and supplementation may prevent its onset (Table 2). P4 levels are lower in the urine and blood of PE patients than in normal control patients during pregnancy [4,11,41,44,45,50,51,52,53,54]. A systematic review and meta-analysis including several RCTs showed that the use of vaginal micronized P4 in the first trimester may reduce the risk of PE (RR, 0.61; 95%CI, 0.41–0.92; 3RCTs, *n* = 5267, moderate-certainty evidence; OR, 0.64; 95%CI, 0.42–0.98; 3RCTs, *n* = 3982, moderate-certainty evidence) in singleton pregnancies [5,6]. However, a retrospective cohort study including 2056 twin pregnancies showed that micronized vaginal P4, or its combination with aspirin, was not superior to aspirin alone in reducing the risk of PE or HDP in twin pregnancies [55]. This evidence showed the beneficial effect of vaginal P4 supplementation in early pregnancy to prevent PE in singleton pregnancies.

Additionally, several previous reviews have shown the potential benefits of dydrogesterone in preventing PE and HDP [56,57]. A comparative retrospective study showed that dydrogesterone supplementation in the first and second periods of pregnancy (from 6 to 20 weeks of gestation) significantly reduced PE incidence [study group (*n* = 169) vs. control group (*n* = 237): 13.1% vs. 71.4%, *p* < 0.001] and reduced the risk of hypertension, proteinuria, FGR, uterus–placenta velocity destruction, and preterm labor in women with higher-risk pregnancies [7]. Furthermore, a clinical case showed that a patient with early sPE during the first pregnancy—to prevent PE in a second pregnancy—received dydrogesterone (30 mg) from 6 to 37 weeks of gestation, and had no clinical signs of PE, including blood pressure <120–130/70–80 mmHg, no proteinuria, no edema, and no fetal or placental abnormalities [8]. These reports indicated that oral dydrogesterone (30 mg, in first and second trimesters) may be the preferred form and dosage for P4 supplementation to prevent PE.

Moreover, in pregnant women with assisted reproductive technology (ART), it is reported that early dydrogesterone supplementation can also decrease PE incidence and outcomes [study group (*n* = 570) vs. control group (*n* = 570): 8.4% vs. 14.2%, *p* < 0.05, a retrospective comparative analysis], HDP and fetal destress [study group (*n* = 116) vs. control group (*n* = 116): 1.7% vs. 12.9%, *p* = 0.001; 4.3% vs. 18.1%, *p* = 0.001; a prospective cross-sectional comparative study], and fetal weight [study group (*n* = 136) vs. control group (*n* = 91): 82.4% vs. 70.3%, *p* = 0.033, secondary data analysis study based on two randomized control trials] [58,59,60]. However, women supplemented with dydrogesterone only showed a lower PE incidence compared with women who received a combination of dydrogesterone and hydroxyprogesterone caproate. This was, nevertheless, not statistically significant [59]. However, it has also been reported that early vaginal P4 supplementation (from embryo transfer to 8 weeks of gestation) in pregnancies with natural cycle frozen embryo transfer (NC-FET) decreased the incidence of PE and HDP, but the difference was not statistically significant [60]. Furthermore, Conrad et al. [61] reported that the risk of developing PE and sPE increased specifically in women undergoing autologous frozen embryo transfer (FET) in artificial cycles without the formation of a corpus luteum relative to natural, modified natural, stimulated, or controlled ovarian stimulation cycles and spontaneous pregnancies. This evidence demonstrated that P4 supplementation is also beneficial for preventing PE in ART.

This evidence suggests that P4 may play an important role in PE pathogenesis and may be involved in its early prediction and prevention. Next, we focused on the possible mechanism by which P4 affects PE progression, mainly by activating nuclear and membrane P4 receptors.

**Table 2 biomolecules-15-01458-t002:** Clinical research showing the potential of P4 support in preventing PE.

PMID	Study Design	Results
37941309 [5]	A systematic review and meta-analysis: Meta-analyses with random-effects model MEDLINE, PubMed, CENTRAL, Embase, and ClinicalTrials.gov (until 20230620) 11 RCTs (3 single center, 8 multiple centers) 11640 patients (5267 patients in 3 RCTs initiated vaginal P4 in the first trimester; 6373 patients in 8 RCTs initiated vaginal P4 in the second and third trimester) Vaginal micronized P4 (100–800 mg/d) Subgroup analysis: 400 mg/bid vs. 400 mg/qd	Vaginal P4 in the first trimester: HDP (RR 0.71, 95%CI 0.53–0.93, 2 RCTs, *n* = 4431, moderate-certainty evidence) Subgroup analysis: a benefit in using the 400 mg/bid regimen (RR 0.74, 95%CI 0.55–0.99, 1 RCT, *n* = 4153) PE (RR 0.61, 95%CI 0.41–0.92, 3 RCTs, *n* = 5267, moderate-certainty evidence) Subgroup analysis: a benefit in using the 400 mg/bid regimen (RR 0.65, 95%CI 0.42–0.99, 2 RCTs, *n* = 4989) Vaginal P4 in the second or third trimesters: HDP (RR 1.19, 95%CI 0.67–2.12, 3 RCTs, *n* = 1602, low-certainty evidence) PE (RR 0.97, 95%CI 0.71–1.31, 5 RCTs, *n* = 4274, low-certainty evidence)
34732210 [6]	A systematic review and meta-analysis: Pairwise meta-analyses with random-effects model MEDLINE, Cochrane Library, Embase and ClinicalTrials.gov (until 20210403) 9 RCTs, 6125 singleton pregnancies Study group: 3055 women treated with P4 before 20 weeks gestation (8 RCTs used oral or vaginal P4; 1 RCTs used rectal P4) Control group: 3070 women unexposed	P4 group: PE (OR 0.64, 95%CI 0.42–0.98, 3 RCTs, *n* = 3982, moderate-certainty evidence) Subgroup analysis: a benefit in using vaginal P4 (OR 0.62, 95%CI 0.40–0.96)
38621482 [55]	A retrospective cohort study: 1800 twin pregnancies managed at a tertiary referral center in the UK (200001–202311) P4 group: 69 treated with P4 only P4+aspirin group: 105 treated with P4 and aspirin in the first trimester Aspirin group: 1156 treated with aspirin only Control group: 470 treated with no medication	PE incidence: Control group vs. Aspirin group vs. P4 group vs. P4+aspirin group: 7% vs. 4.6% vs. 5.8% vs. 7.6% HR for PE: Aspirin group: 0.62, 95%CI 0.40–0.97, *p* = 0.036 P4 group: 0.83, 95%CI 0.37–1.83, *p* = 0.637 P4+aspirin group: 0.71, 95%CI 0.25–2.03, *p* = 0.527
31876197 [7]	Comparative retrospective study: 406 pregnancies with risk factors of PE Study group: 169 pregnancies with dydrogesterone supplementation (30 mg/d) at 6–20 weeks gestation Control group: 237 pregnancies without dydrogesterone supplementation	PE: 13.1% and 71.4%, *p* < 0.001 Hypertension: 3.2% and 71.2%, *p* < 0.001 Proteinuria: 0.0% vs. 66.18%, *p* < 0.001 FGR: 2.2% vs. 21.58%, *p* < 0.001 Destroy of uteri-placenta velocity: 3.2% vs. 21.58%, *p* < 0.001 PL: 8.6% vs. 53.95%, *p* < 0.001
39722356 [60]	A secondary data analysis study based on 2 randomized control trials (2008–2011 and 2013–2018) at 2 university hospitals in Sweden 227 singleton pregnancies of NC-FET Study group: 136 pregnancies with luteal phase vaginal P4 supplementation from ET to 8 weeks gestation 56 pregnancies, 2008–2011, micronized P4, 400 mg/bid, as a vaginal suppository 80 pregnancies, 2013–2018, P4, 100 mg/bid, as a vaginal tablet Control group: 91 pregnancies, receive standard of care, without P4 supplementation	Study group vs. Control group: Birth weights: 82.4% vs. 70.3%, *p* = 0.033 HDP: 4.4% vs. 11.1%, *p* = 0.058 Mild/moderate PE: 1.5% vs. 3.3% Severe PE: 1.5% vs. 2.2%
24552449 [58]	A prospective cross-sectional comparative study: 232 primigravidae in a tertiary center (201001-201012) Study group: 116 primigravidae with dydrogesterone supplementation (10 mg/tid) following ART or IUI up to 16 weeks gestation Control group: 116 primigravidae, age and race matched spontaneous pregnancies at 16 weeks, without dydrogesterone supplementation	Study group vs. Control group: HDP: 1.7% vs. 12.9%, *p* = 0.001 fetal distress: 4.3% vs. 18.1%, *p* = 0.001
26910749 [59]	A retrospective comparative analysis: 1140 pregnancies in a tertiary center (200601–201503) Study group: 570 with P4 support following ART or IUI until 14–16 weeks gestation Control group: 570 without P4 support	Study group vs. Control group: PE: 8.4% vs. 14.2%, *p* < 0.05 Subgroup analysis: Dydrogesterone only (10 mg, *n* = 276) vs. Dydrogesterone+hydroxyprogesterone (intramuscular injection of 500 mg, *n* = 294): PE: 6.9% vs. 9.9%, *p* = 0.2

## 4. P4’s Genomic and Non-Genomic Mechanisms of Action

### 4.1. P4 Classical Signaling Pathway: Genomic Receptor Mechanism (Nuclear)

P4 is a lipophilic molecule that readily crosses cell membranes by diffusion and interacts at the nuclear level with nuclear P4 receptors (nPGRs), PGR-A (94 kDa), and PGR-B (120 kDa) [62]; thus, this activates co-regulators that act on ribosomal RNA, resulting in the production of corresponding proteins [63,64]. Notably, higher P4 levels downregulated PGR levels, which in turn induced further P4 secretion [65]. A feedback loop exists between P4 and PGR. The unbound receptor exists in both subcellular compartments and resides in complexes with heat-shock protein (HSP) chaperone molecules, such as HSP70 and HSP90 [66]. Ligand binding to PGR induces dissociation from HSPs, the retention of dimerized PGR in the nucleus, and transcription activation. The genomic receptors PGR-A and PGR-B are encoded by a single gene but are transcribed from different promoters to generate distinct subclasses of PGR mRNAs [67]. These PGR-A and PGR-B isoforms share the transcription-activating function (AF) domains AF-1 and AF-2; the PGR-B isoform contains a specific N-terminal segment, AF-3, as well as the B-upstream segment that is absent from the PGR-A isoform. Both PGR-A and PGR-B contain a C-terminal ligand-binding domain and hinge region [64].

Since there are various cell types at the maternal–fetal interface, including trophoblasts, immune cells, and decidual stromal cells, we first discuss the distribution of PGRs in humans and other mammalian species. In human and primate placentas, PGR is expressed in stromal cells, spiral arteries, and/or the myometrium from early to late pregnancy [68]. However, whether PGR is expressed in trophoblasts remains controversial. Research has shown that PGR-A/B is also expressed in trophoblasts, as well as spatial differences in PGR-B dominance in early and PGR-A in late trophoblasts from legal abortion [68]. PGRs are expressed in several trophoblast cell lines, such as HTR-8/SVneo, JEG3, and BeWo cells [69,70]. However, research has also shown that JEG3 and EVTs from primary trophoblasts do not express PGR for P4 [71,72,73,74]. In other mammals, the PGR expression range differs. In canines, decidual stromal cells are the only placental cell population that expresses PGR [75], which has been detected at the mRNA and protein levels in several placental compartments during early pregnancy in sheep [76,77]. In pigs, PGR is expressed in the uterine stroma and myometrium throughout pregnancy [78]. Due to the varying PGR expression levels in different cell types, P4 may affect different cell types, thereby having different impacts on cell functions and overall maternal–fetal interface function.

Research showed that the abnormal expression of PGR in the placenta is associated with PE development (Table 3). It is reported that PGR-B gene expression and protein abundance are remarkably disrupted in the decidua of sPE pregnancies [79]. Transcription factors PGR and cAMP-responsive element binding protein 1 (CREB1) jointly regulate the transcription of downstream genes in response to decidual signals, influencing the decidualization of endometrial stroma [80]. Additionally, it has been reported that PGR levels were elevated in preeclamptic placentas [70], and that this can influence trophoblast function. For example, P4 can promote trophoblast cell proliferation and invasion [44], and PGR deficiency downregulates cyclin D1 protein levels in trophoblasts, which plays an important role in cell proliferation [65]. This evidence indicates that P4 influences PGR expression at the maternal–fetal interface, thus leading to PE development.

### 4.2. P4 Non-Classical Signaling Pathway: Non-Genomic Receptor Mechanism (Extranuclear)

#### 4.2.1. mPRs

Specific membrane-bound P4 receptors (mPRs), also called progestin and adipoQ receptors (PAQRs), have been described as having seven transmembrane domains, similar to G protein-coupled receptors [88]. There are currently five mPR subtypes (mPRα, mPRβ, mPRγ, mPRδ, and mPRε). Initial studies have indicated that mPR expression is highly tissue-specific, with mPRα being the predominant isoform in reproductive tissues, mPRβ in neural tissues, and mPRγ in the gastrointestinal tract. After P4 activates mPRs, it can affect the biological functions of cells by activating downstream cascade reactions, including stimulating extracellular signal-regulated kinases 1/2 (ERK1/2 or p42/44), p38 mitogen-activated protein kinase (MAPK), or intracellular Ca^2+^ mobilization [64].

Research has shown that mPRα is highly expressed in the placenta, and that the tissue expression levels of mPRα genes are negatively correlated with PGR expression, suggesting important functional roles for mPRα in specific reproductive tissues, particularly those that express low levels of nuclear PGR [86]. P4 stimulates mPRα and involves signaling through the phosphatidylinositol 3-kinases/protein kinase B (PI3K/AKT) and MAPK pathways to produce nitrogen oxide (NO) in human umbilical vein endothelial cells (HUVECs); this is important in vasodilatation [87]. Additionally, P4 binds to mPR in macrophages and upregulates the mRNA expression of cyclooxygenase 2 (COX2), TNF-α, and prostaglandin-endoperoxide synthase 2 (PTGS2), leading to the activation of the MAPK and protein kinase A (PKA) pathways. This thus contributes to the inflammatory responses in macrophages with subsequent labor regulation [89]. The above content is presented in Table 3.

#### 4.2.2. PGRMC

Another membrane-bound P4 receptor, P4 receptor membrane component (PGRMC), was isolated from neural tissue and is distinct from known mPRs and nPGRs [90]. Within this protein family, PGRMC1 and PGRMC2 are the best characterized, as they mediate some of P4’s actions. Additionally, PGRMCs have both P4-dependent and -independent functions, and may more broadly serve as promiscuous receptors for selecting sterols [91,92]. This is supported by recent findings in breast cancer cell lines showing that different synthetic progestins cause many of the same cellular outcomes when signaling through the classical or nuclear P4 receptor, but have seemingly distinct PGRMC1-dependent actions [93,94]. For example, medroxyprogesterone acetate, norethisterone, and dienogest—but not nomegestrol acetate—induce proliferation in MCF7 and T47D breast cancer cells through a mechanism dependent on the phosphorylation of a casein kinase 2 domain within PGRMC1 [94,95,96].
PGRMC1

PGRMC1 plays a crucial role in placental function (Table 3). Research has shown that PGRs are absent in EVTs and JEG3, whereas PGRMC1 and PGRMC2 are highly expressed in all trophoblasts [71,72,73,74]. It has been reported that JEG3 invasion is inhibited by P4 antagonist RU486 (mifepristone) in a PGRMC1-dependent manner [74]. Additionally, PGRMC1 downregulation accelerates the differentiation and fusion of BeWo cells [81]. In addition, it significantly promotes differentiation, decidualization, and cellular senescence in endometrial stromal cells (ESCs) [82,83].
PGRMC2

PGRMC2 may also affect placental function (Table 3). Its attenuation plays a role in placentation by promoting cell proliferation, invasion, and angiogenesis in EVTs via the activation of hypoxia-inducible factor 1 alpha (HIF-1α) signaling [84]. An organ-on-chip model of the maternal–fetal membrane interface revealed that HLA-G and PGRMC2 produced by chorionic trophoblasts modulate inflammation, thus serving as an immunological barrier preventing membrane compromise [97]. Furthermore, PGRMC2 knockdown significantly compromised the ability of decidualized ESCs to support trophoblast expansion in an outgrowth model [85].

The above evidence suggests that P4 exerts its downstream functions at the maternal–fetal interface, mainly through activating nuclear and membrane P4 receptors. Next, we review the effects of P4 and the activation of its receptors on cellular function at the maternal–fetal interface, focusing primarily on three aspects: vascular function, immune response, and placental function.

## 5. P4 Might Protect Against PE by Regulating Vascular Endothelial Function

Vascular endothelial dysfunction is a typical characteristic of PE, characterized by oxidative stress and inflammatory response [98]. Research has shown that P4 plays an important role in vascular remodeling by regulating vascular endothelial function [99,100,101], thus likely contributing to its protected role in PE.

### 5.1. P4 Might Protect Against PE by Regulating VEGF Signaling

Vascular endothelial growth factor (VEGF) has emerged as a significant biomarker in the context of PE. Research indicates that altered levels of VEGF can be associated with the onset and severity of this condition, making it a focal point for both diagnostic and prognostic assessments [102,103].

P4 plays an important role in protecting vascular endothelial function in vascular remodeling by regulating VEGF signaling; it is thus involved in PE (Figure 3). On one hand, P4 levels are correlated with VEGFA expression. It has been reported that the glandular epithelial expression of VEGFA, VEGFC, and placental growth factor (PlGF) is higher in women with elevated P4 levels than in those with normal levels. Moreover, a significantly higher stromal expression of VEGFA and PLGF was found in women with elevated P4 levels [104]. On the other hand, P4 promotes VEGF expression to protect vascular endothelial function. Previous research has shown that ovarian stimulation and P4 administration enhance endometrial angiogenesis by upregulating VEGF expression [105,106]. Additionally, research has shown that P4 governs uterine angiogenesis and vascular remodeling via VEGFA/VEGFR2 signaling, especially in the anti-mesometrial region where the embryo resides during pregnancy [107,108,109]. Moreover, P4 alleviates dexamethasone-induced intrauterine growth retardation (IUGR), probably by promoting placental VEGF and angiogenesis [110]. Furthermore, in PE, P4 blunts the vascular endothelial cell secretion of endothelin-1, a partner of VEGF, in response to placental ischemia [41].

Notably, P4 is involved in regulating VEGF signaling and vascular endothelial function, which is dependent on the P4 receptor (Figure 3). P4 improves the angiogenic potential of endothelial progenitor cells, including tube formation, adhesion, migration, and VEGF secretion, in a dose-dependent manner, which can be reversed by a P4 receptor antagonist [111]. In addition, PGRMC1 overexpression significantly promoted the cellular processes associated with endothelial cell proliferation, migration, and angiogenesis [112]. Additionally, high mPRα expression enhances the activation of cAMP-Janus kinase-signal transducer and activator of transcription (JAK/STAT) signaling and increases HIF-1α-induced VEGF secretion into the tumor microenvironment, promoting HUVEC migration and tube formation under hypoxia in lung adenocarcinoma [113].

This evidence showed that, by regulating VEGF signaling and vascular remodeling, P4 and its receptor might be involved in PE development.

### 5.2. P4 Might Protect Against PE by Regulating eNOS/NO Expression

Preeclampsia (PE) is a complex pregnancy disorder characterized by hypertension and proteinuria [1]. One of the critical pathways implicated in the pathophysiology of PE is the endothelial nitric oxide synthase (eNOS)/nitric oxide (NO) signaling pathway [114,115]. Changes in this pathway can lead to impaired endothelial function, contributing to the vascular complications associated with PE [114,115]. Research indicates that eNOS expression and activity are significantly altered in preeclamptic patients, leading to reduced NO bioavailability. This reduction in NO is associated with increased vascular resistance and hypertension, hallmark features of preeclampsia [114,115].

P4 promotes NO production to protect vascular endothelial cells and is thus involved in PE (Figure 3). For example, P4 promotes NO production, inhibiting cellular antioxidant effects and increasing oxidative stress in arterial endothelial cells [116]. Additionally, P4 stimulates NO production in HUVECs and is involved in the PI3K/AKT and MAPK pathways by regulating by mPRα [87,117].

P4 promotes NO production by increasing eNOS expression. Studies have shown that P4 promotes vasodilation by regulating transcription factor specificity protein 1 (SP1) in endothelial cells, enhancing eNOS expression and further increasing NO synthesis [118]. Importantly, P4 may be important in increasing eNOS expression through mPRα and subsequent PI3K/AKT, as well as MAPK pathways, exerting beneficial vascular function effects [87,119,120,121]. This evidence showed that P4 and its receptor might be involved in PE development via regulating eNOS and NO expression.

In the above discussion, P4 might protect against PE by regulating vascular endothelial function, as well as VEGF signaling and eNOS and NO expression.

## 6. P4 Might Protect Against PE by Regulating Immune Response at the Maternal–Fetal Interface

Immune dysfunction at the maternal–fetal interface is also a key factor involved in PE pathophysiology. On one hand, immune cells at this interface in patients with PE tend to convert to a pro-inflammatory phenotype and secrete pro-inflammatory cytokines, accelerating PE onset and progression. Studies have found that, in patients with PE, there is an increase in the Th1/Th2 ratio [122,123,124], pro-inflammatory M1-type macrophages [125,126,127,128], cytolytic NK cells [129,130,131], and the pro-inflammatory cytokines TNF-α and interferon-gamma (IFN-γ), alongside a decrease in Treg cells [132,133], mature dendritic cells (DCs) [134,135,136], and anti-inflammatory cytokines. These changes lead to impaired immune suppression at the maternal–fetal interface and impaired uterine spiral artery remodeling, contributing to PE development. On the other hand, in patients with PE, human leukocyte antigen-C (HLA-C) and HLA-G expression is reduced in trophoblasts [137,138], which impairs their immunosuppressive function on immune cells, undermining immune tolerance at the maternal–fetal interface and further exacerbating PE development.

Research has shown that P4 plays an important immunomodulatory role during pregnancy, mainly promoting immune tolerance at the maternal–fetal interface by influencing immune cell [139,140] and trophoblast function, and thus protecting against PE. In addition, in the presence of P4, PGR-positive pregnancy lymphocytes produce a protein called P4-induced blocking factor (PIBF), which mediates some of P4’s immunological effects [141,142].

### 6.1. P4 Might Protect Against PE by Regulating Immune Cells at Maternal–Fetal Interface

#### 6.1.1. Th1/Th2

P4 can promote PIBF production, which inhibits the Th1/Th2 ratio at the maternal–fetal interface (Figure 4). In humans, P4-treated human pregnancy lymphocytes produce PIBF, followed by decreased Th1-type cytokine and increased Th2-type cytokine production, which can promote T cell differentiation into Th2 [143,144]. Similarly, PIBF-treated spleen cells from non-pregnant female mice produce significantly more Th2-type cytokines, interleukin-4 (IL-4) and interleukin-10 (IL-10), than those without PIBF [145]. Furthermore, T cells from PIBF-deficient pregnant mice differentiate into Th1 [146].

P4 protects against PE by inhibiting the Th1/Th2 ratio at the maternal–fetal interface. For example, PIBF treatment normalized the Th1/Th2 ratio, reduced inflammation, corrected blood pressure, and prevented FGR in a rat model of PE [147].

#### 6.1.2. Treg Cells

P4 promotes Treg cells at the maternal–fetal interface (Figure 4). Research has shown that P4 supports Treg expansion and suppressive function at this interface during pregnancy, as well as a skew toward an anti-inflammatory cytokine profile and CD8+T cell cytotoxicity suppression [49,139,148,149,150,151,152]. Impaired luteal phase P4 signaling following the administration of the P4 antagonist RU486 (mifepristone) causes Treg cell deficiency in a mouse model [49].

P4 might protect against PE by expanding the number of Treg cells at the maternal–fetal interface. In mice, PGR deletion on DCs promotes a non-tolerogenic, mature DC phenotype, along with failure to generate CD4+Tregs and CD8+CD122+Treg cells and impaired placental and fetal development [151].

#### 6.1.3. CD8+T Cells

P4 decreases the number of CD8+T cells at the maternal–fetal interface (Figure 4). It has been reported that vaginal P4 decreases the proportion of decidual CD8+CD25+Foxp3+T cells at this interface in a mouse model [152]. Therefore, we suppose that P4 might protect against PE by decreasing the number of CD8+T cells at this site.

#### 6.1.4. Macrophage and DCs

P4 promotes a tolerogenic profile of macrophages and dendritic cells (DCs) (Figure 4). For example, in vitro stimulation with P4 induces the maturation of macrophages with an M2-type profile [139,153,154] and prevents dendritic cell differentiation into a mature phenotype [155]. In addition, it has been reported that vaginal P4 decreases the proportion of decidual macrophages at the murine maternal–fetal interface [152].

P4 promotes a tolerogenic profile of macrophages and DCs, which are essential for successful uterine tissue remodeling, pregnancy maintenance, and reducing pregnancy complications such as PE [134,135,136]. Recent evidence has revealed that, in mice, the targeted deletion of PGR on DCs promotes a non-tolerogenic, mature DC phenotype, along with failure to generate CD4+Tregs and CD8+CD122+Treg cells and impaired placental and fetal development [151].

#### 6.1.5. NK Cell

P4 can promote PIBF production, which inhibits dNK cell activity at the maternal–fetal interface (Figure 4). Research has shown that the PIBF present in the cytoplasmic granules of dNK cells contributes to the low dNK activity by inhibiting the release of perforin and other cytotoxic molecules [156].

P4 protects against PE by inhibiting dNK cell activity at the maternal–fetal interface. PIBF has been shown to decrease inflammation and cytolytic NK cells to prevent hypertension in a RUPP rat model of PE [47]. The adoptive transfer of spleen cells with high NK activity to pregnant mice increases fetal loss, which is counteracted by PIBF treatment [157]. In mice, PIBF depletion during the peri-implantation period results in reduced implantation and increased resorption rates, together with increased decidual and peripheral NK cell activity [146]. By contrast, the increased resorption rates observed in PIBF-depleted mice were corrected by treating them with anti-NK antibodies [158], suggesting that PIBF contributes to successful murine gestation by controlling NK activity.

#### 6.1.6. Cytokine

In addition, P4 promotes the production of anti-inflammatory cytokines while inhibiting that of pro-inflammatory cytokines, thus hindering the immune response at the maternal–fetal interface [142,154]. P4 and its metabolites reduce TNF-α and IFN-γ production in CD4+ and CD8+T cells [159], which are important for immune tolerance at the maternal–fetal interface. Additionally, P4 can enhance the immunosuppressive environment by upregulating specific cytokines such as IL-10, which is crucial for protecting the fetus from maternal immune attacks [154].

P4 protects against PE by regulating pro- and anti-cytokines at the maternal–fetal interface. For example, in PE women, after P4 administration, TNF-α and interleukin-1beta (IL-1β) levels were decreased, while those of IL-4, IL-10, interleukin-13 (IL-13), cyclin D1, and proliferating cell nuclear antigen (PCNA) were increased [44]. P4 can significantly reduce interleukin-18 (IL-18) expression and increase CD56+CD16-NK cell growth, suggesting that these activities may underlie the mechanism by which P4 improves pregnancy outcomes [160]. Moreover, P4 treatment significantly reduced the secretion of inflammatory cytokines in the serum, macrophages, and trophoblasts of B. abortus-infected mice, leading to decreased placentitis and enhanced pup viability [161]. Insufficient P4 may lead to the activation of inflammatory responses in the placenta, thereby triggering PE [44].

Therefore, P4 can inhibit the activation of various pro-inflammatory cells (e.g., Th1 cells, CD8+T cells, M1-type macrophages, and cytotoxic NK cells) and cytokines (e.g., TNF-α and IFN-γ), and increase anti-inflammatory cells (e.g., Th2 cells, M2-type macrophages, and Treg cells) and cytokines (e.g., IL-10 and IL-4) at the maternal–fetal interface, exhibiting a net anti-inflammatory effect supporting successful pregnancy.

### 6.2. P4 Might Protect Against PE by Regulating Trophoblasts for Immune Tolerance

P4 stimulation has been reported to enhance the expression of HLA-C and HLA-G, which are key molecules involved in immune tolerance [74]. An organ-on-chip model of the maternal–fetal membrane interface revealed that HLA-G and PGRMC2 produced by chorionic trophoblasts modulate decidual immune cell inflammation, thus serving as an immunological barrier preventing membrane compromise [97]. In addition, in the placenta, P4 not only regulates immune cell activity, but also influences the secretory function of placental cells and regulates the placental response to the maternal immune system, thereby ensuring smooth pregnancy progression [162]. P4-driven B7-H4 (VTCN1), an immunosuppressive protein in trophoblasts, contributes to maternal–fetal immune tolerance by inhibiting CD8+T cell activity [163]. This evidence demonstrates the effect of P4 on trophoblasts for immune tolerance, which might influence immune cell function to protect against PE (Figure 4).

## 7. P4 Might Protect Against PE by Promoting Trophoblast Proliferation and Adhesion

Trophoblast cells, the primary cell type in the placenta, are crucial for successful implantation and placental development. In normal pregnancies, trophoblasts undergo a process of differentiation and invasion that allows them to remodel maternal blood vessels and establish a healthy placental environment. However, when these processes are disrupted, preeclampsia develops, leading to inadequate placental ischemia and elevated blood pressure in the mother [164,165,166,167]. The ability of trophoblasts to proliferate and adhere is essential for their invasive capabilities, and any impairment in these functions can trigger the cascade of events that culminate in preeclampsia [168,169].

Research has shown that P4 treatment can promote trophoblast proliferation and adhesion and inhibit trophoblast apoptosis [44,46,170]. P4 upregulates protein O-fucosyltransferase 1 (poFUT1) expression via the specific transcription factor AP-1 family members (c-Fos/c-Jun) and then increases trophoblast cell proliferation and adhesion potential [170]. Research has also shown that PGR deficiency downregulates cyclin D1 protein levels, which plays an important role in JEG3 cell proliferation [65]. In mouse trophoblast giant cells (TGCs), P4 protects against lamin B1 loss and prolongs the life and function of TGCs [171]. This evidence indicated that P4 might protect against PE by regulating proliferation and adhesion (Figure 5).

## 8. Conclusions

In this review, we summarize the clinical implications of progesterone in preeclampsia. Current RCT studies, systematic reviews, and meta-analyses suggest that P4 supplementation has a preventive effect on PE incidence in patients with singleton pregnancies, both in normal conception and assisted reproductive technology. On one hand, key enzyme or metabolite abnormalities during P4 synthesis and metabolism lead to PE development. On the other, P4 might protect against PE by regulating immune response at the maternal–fetal interface, inhibiting vascular function, and diminishing trophoblast proliferation and adhesion. However, research should also focus on the comprehensive mechanisms by which P4 is involved in PE pathogenesis at the maternal–fetal interface. Additionally, despite preliminary positive results, large-scale randomized controlled trials are required to validate the efficacy and safety of P4 interventions. Future research should focus on optimal administration routes, dosages, and the combined use of P4 with other therapeutic approaches to provide more effective strategies for managing gestational hypertension.

## Figures and Tables

**Figure 1 biomolecules-15-01458-f001:**
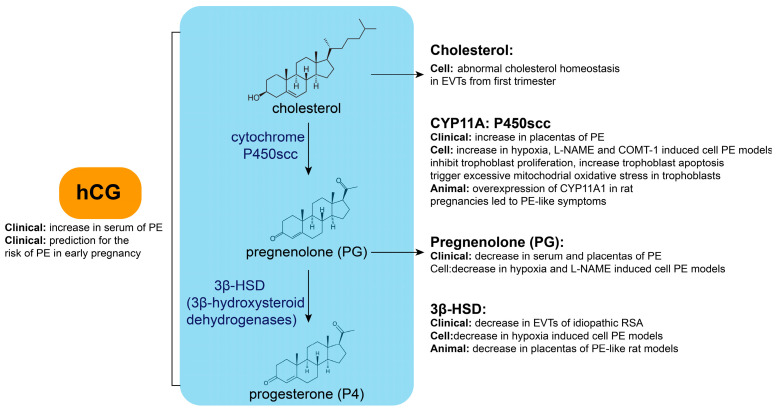
Key enzyme or metabolite abnormalities during P4 synthesis in PE.

**Figure 2 biomolecules-15-01458-f002:**
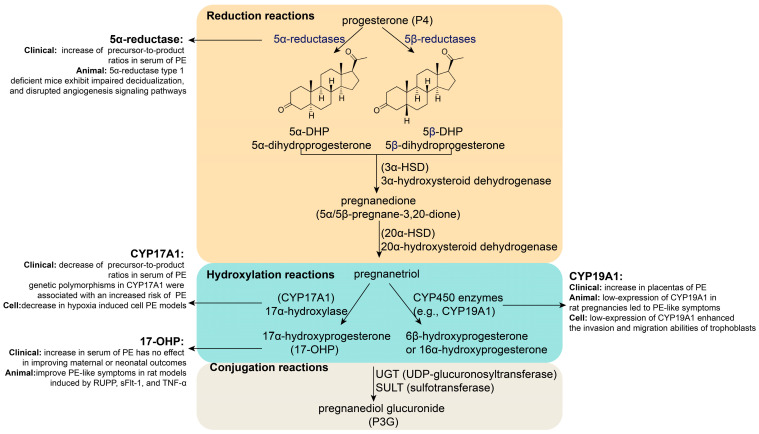
Key enzyme or metabolite abnormalities during P4 metabolism in PE.

**Figure 3 biomolecules-15-01458-f003:**
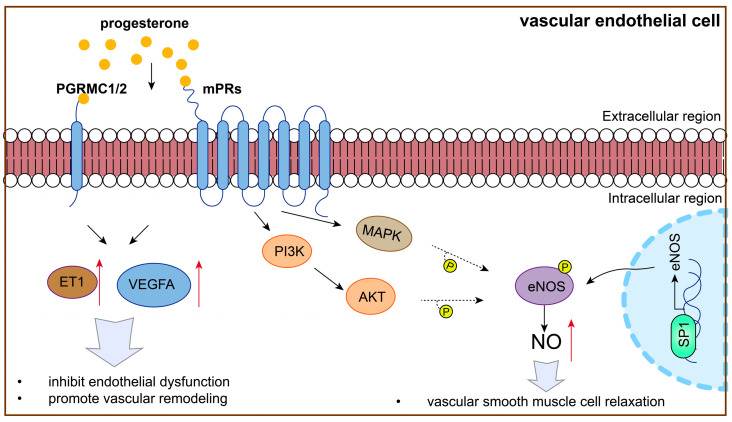
P4 might protect against PE by regulating vascular endothelial function.

**Figure 4 biomolecules-15-01458-f004:**
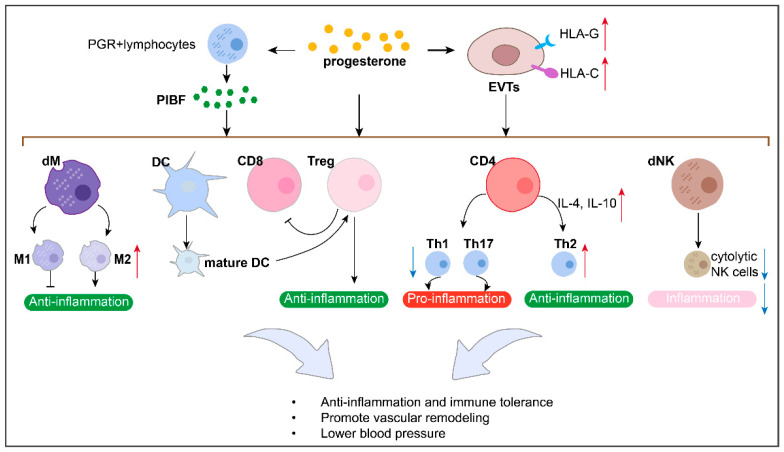
P4 might protect against PE by regulating immune response in maternal–fetal interface.

**Figure 5 biomolecules-15-01458-f005:**
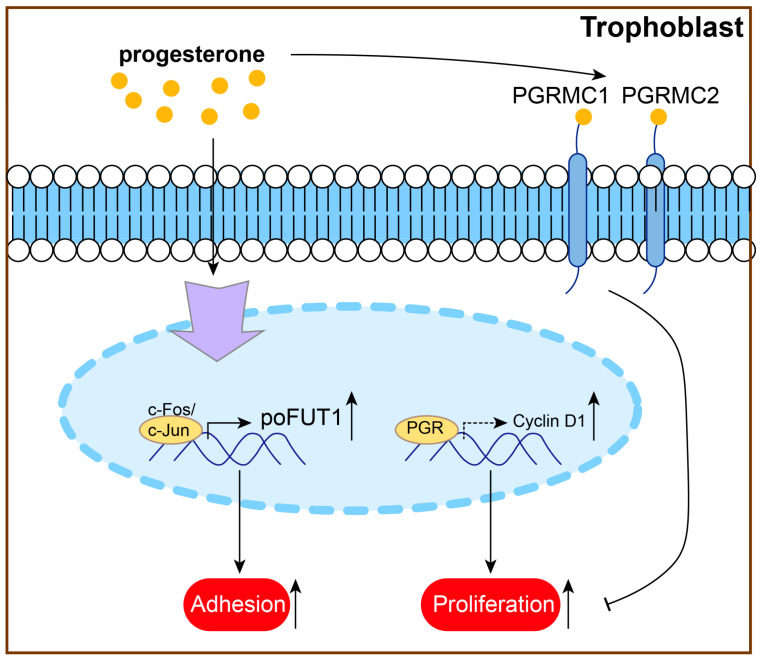
P4 might protect against PE by promoting trophoblast proliferation and adhesion.

**Table 1 biomolecules-15-01458-t001:** Animal research showing the potential of P4 support in preventing PE.

PMID	Animal Models	Symptoms	Treatment	Results
29405962 [4]	PE-like rat model: intraperitoneally administrating cadmium (0.125 mg/kg) on GD9-12	hypertension, proteinuria and placental abnormalities	Intraperitoneal injection with P4 (3 mg/kg) from GD9 to delivery	The symptoms were improved after treatment with P4
36593749 [44]	PE-like rat model: subcutaneously injecting L-NAME (125 mg/kg/d) from GD12 for 7 days	hypertension, proteinuria, and downregulation of MMP2 and MMP9 in serum	Oral treatment with P4 (0, 10^−8^, 10^−6^, 10^−4^) daily from GD9	The symptoms were improved after treatment with P4
35327614 [46]	Systematic SIRT1^+/−^ mice:	hypertension, proteinuria, FGR, kidney injury, placental injury	Intraperitoneal injection with P4 (3 mg/kg) daily at GD7.5-18	The symptoms were improved after treatment with P4
33533305 [47]	PE-like rat model: Performing RUPP on GD14	hypertension, FGR, and inflammation (Circulating and placental cytolytic NK cells, IL-17, and IL-6 increased while IL-4 and Th2 cells decreased)	Intraperitoneal injection with PIBF (2.0 μg/mL) at GD15	The symptoms were improved after treatment with PIBF
38282604 [48]	PE-like rat model: intraperitoneally injecting rabbit anti-PIBF IgG (0.25 mg/mL; 0.50 mg/mL) on GD15	MAP elevated, cytolytic NK cells and TNF-α in plasma increased, IL4 and IL10 in plasm decreased	——	——
37191999 [49]	Model luteal phase P4 deficiency: Injecting P4 antagonist RU486 (0.5–8 mg/kg) on GD0.5 and GD3.5	Fetal loss and FGR, impaired Treg number and function	——	——

**Table 3 biomolecules-15-01458-t003:** The expression and function of several P4 receptors.

Receptor Type	Expression in Placenta	The Relationship with PE	Function in Trophoblast	Function in ESCs
PGR	stromal cells, spiral arteries, myometrium [68]; whether PGR is expressed in trophoblasts is controversial [68,69,70,71,72,73,74]	Increase in placentas of PE [70]	Promote cyclin D1 in JEG3 cells [65]	Promote decidualization of ESCs [80]
PGRMC1	highly expressed in placenta [71,72,73,74]	——	Promote invasion of JEG3 cells [74] Inhibit differentiation and fusion of BeWo cells [81]	Inhibit differentiation, decidualization, and senescence of ESCs [82,83]
PGRMC2	highly expressed in placenta [71,72,73,74]	——	Promote proliferation, invasion, and angiogenesis of EVTs by activating HIF-1α signaling [84]	Promote decidualization of ESCs to support trophoblast expansion [85]
mPRα (PAQR7)	highly expressed in the placenta [86]	Product NO in HUVECs for vasodilatation by activating MAPK and PI3K/AKT pathways [87]	——	——

## Data Availability

Not applicable.

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
