# Peer review of "The Clinical Implications of Progesterone in Preeclampsia"

_biomolecules, 2025, doi:10.3390/biom15101458_

Round 1
Reviewer 1 Report
Comments and Suggestions for Authors
Although the content of this manuscript is not very much different from a recent review (DOI: 10.3390/ijms23031333) which seemed to address a similar problem, it is needed in the field. The authors have demonstrated some form of understanding in how review articles are written. For instance, the introduction section highlights the aim of the review and how the review was done. Also, at each stage of the review, the authors tried to make deductions, which reflect a good review manuscript. Regardless of this, I have some issues with this manuscript which I believe, if addressed, will improve its quality.
- Minor grammatical errors are seen in the abstract and the entire manuscript, and should be rechecked.
- While the clinical implications of P4 in PE have been clarified in this review, the pathophysiological role of P4 in PE has not been substantiated. Hence, the title of this manuscript is misleading.
- In line 43, “determine” should be replaced with “characterize” since this is not a research article but a review of available findings to make a claim.
- Some unnecessary paragraphing are also seen in this manuscript. For instance, the paragraph created at line 74 is a mistake. From line 62 to 82, the authors were trying to synthesize a common information. So, no paragraphing is needed there. Authors should note that in scientific writing, a deduction made at the end of a paragraph is about that paragraph not about other paragraphs. So, the deductions they made in line 81 will be mistaken for only the information presented from lines 74 to 79. However, its obvious that it is also applicable to the preceding paragraph. The entire manuscript should be devoid of such paragraphing anomalies.
- In line 263, the authors wrote the heading “P4 protects against PE by regulating vascular endothelial function”. This is a serious claim which, unfortunately, is not supported by the references cited. None of the references is about PE.
- In line 268, the authors wrote a bold statement that “P4 protects against PE by regulating VEGF signaling”. Surprisingly, none of their references supports this claim, and still the authors had the courage to propose a mechanism about this. Wow! While the references point to a potential role of P4 in VEGF signaling, none of the studies was about PE. The same problem is associated with the heading “P4 protects against PE by regulating eNOS/NO expression” and the references cited in its support.
- In line 433, the authors wrote that “P4 protects against PE by regulating trophoblast invasion”. To substantiate this bold claim, they cited papers with opposing findings. In those studies, cell lines, such as HTR8/SVneo and JEG3, which do not closely mimic trophoblast development were employed. So, even if the findings had been consistent across the studies, the results could still not be extrapolated to the ideal process of placentation in humans. Since those cell lines do not yield reliable experimental results and besides the findings are conflicting, the conclusion of the authors is dangerously misleading. They must consider deleting that paragraph or rephrasing everything into a hypothetical message and requesting prospective studies to address that gap in knowledge. It is premature to make any meaningful deduction since the available evidence is not enough.
- In line 449, the authors wrote the heading “P4 protects against PE by promoting trophoblast proliferation and adhesion”. While the findings support the potential role of P4 in trophoblast proliferation, none of the studies was done in relation to PE. The role of P4 in trophoblast adhesion could also not be clarified with enough evidence. What authors can do is to also explain how proliferation and adhesion associate with PE, and then cite the ideal references. Until this is done, the statement made in the heading cannot be seen to be correct.
- From line 462 to 522, the authors reviewed the association between P4 supplementation and PE development. They first presented findings about humans and then moved on to findings about animals. In scientific reports, in vitro studies are first reported, followed by animal studies and then human studies because once findings about humans are concluded, findings from in vitro and animal studies are no longer relevant. In this review, it would have been ideal to present the findings about animals before the findings about humans. Truly, even if the findings about animals are deleted, it won’t change anything. After all, the authors have enough human studies to substantiate their claim. Apart from this, it would have been ideal to present these clinical findings ahead of the mechanistic findings that began from line 263. To the best of my knowledge, association findings are first reported before mechanistic findings are brought in to clarify the underlying pharmacological mechanisms.
- Due to the issues I have already raised, the conclusions of this review cannot be trusted. Authors are to rewrite the manuscript to reflect the hypothetical stance of their message and then revise the conclusions in a similar way.
- My final words to the authors is that in literature review, you don’t have to only look at the findings of a study but also the methodology used. If a research work has a questionable methodology, the findings should not be taken seriously as that can affect the deductions made in a review.
The grammar is good but not excellent. Grammar check is needed to improve the overall quality of the manuscript.
Reviewer 2 Report
Comments and Suggestions for Authors
This is a very interesting scientific study on the role of progesterone in physiological pregnancy and preeclampsia, which is often underestimated: its synthesis, and metabolic abnormalities play important roles in the pathophysiology of preeclampsia. The structure of the study makes the role of the pregnancy hormone clear, as well as where and how it acts in protecting against the disease. The biochemical actions provide a good explanation, giving a comprehensive understanding of the current state of knowledge with effective schematic diagrams. The conclusions are important in resolving doubts and possibilities that therapeutic supplementation in preeclampsia or prophylaxis could be a clinical use to be standardized in clinical practice.
The application of the work could also clarify how its concentration in the placenta recommends, for example, vaginal administration, a route known to have a therapeutic advantage.
The work may give rise to the idea and the need to carry out a randomized multicenter study to evaluate its actual clinical efficacy.
I appreciate this study.
Reviewer 3 Report
Comments and Suggestions for Authors
The article is generally well-written and organized, and addresses an interesting topic. I found the figures particularly well-designed and informative. My only suggestion would be regarding the way the conclusion is written. The way the conclusions are worded makes it unclear whether the effect was protective or not. This is particularly true in the following sentences: "as well as its involvement in the condition’s pathogenesis. On the one hand, we propose that P4 may contribute to PE development by regulating vascular function, placental function, and immune response at the maternal–fetal interface by activating nuclear and membrane P4 receptors." and " This evidence suggests that P4, its synthesis, and metabolic abnormalities play important roles in the pathophysiology of preeclampsia." I suggest revising them in this regard. This would help make the conclusions of the work clearer.
Comments on the Quality of English LanguageMy only suggestion would be regarding the way the conclusion is written. The way the conclusions are worded makes it unclear whether the effect was protective or not. This is particularly true in the following sentences: "as well as its involvement in the condition’s pathogenesis. On the one hand, we propose that P4 may contribute to PE development by regulating vascular function, placental function, and immune response at the maternal–fetal interface by activating nuclear and membrane P4 receptors." and " This evidence suggests that P4, its synthesis, and metabolic abnormalities play important roles in the pathophysiology of preeclampsia." I suggest revising them in this regard. This would help make the conclusions of the work clearer.
Round 2
Reviewer 1 Report
Comments and Suggestions for Authors
The revised manuscript looks better. However, I have one minor issue, which is stated below:
In line 510, the authors wrote “However, this process is disrupted in preeclampsia, leading to inadequate trophoblast invasion and impaired remodeling of spiral arteries, which in turn results in placental ischemia and elevated blood pressure in the mother”.
My message to the authors is that inadequate invasion and spiral artery remodeling rather leads to preeclampsia but not what they have written, so I will suggest they use the sentence below:
“However, when these processes are disrupted, inadequate placental ischemia and elevated blood pressure occur in the mother, and these are major manifestations of preeclampsia”.
